# A Visual Case Study of the Training Dynamics in Neural Networks

## Abstract

This paper introduces a visual sandbox designed to explore the training dynamics of a small-scale transformer model, with the embedding dimension constrained to $d = 2$. This restriction allows for a comprehensive two-dimensional visualization of each layer's dynamics. Through this approach, we gain insights into training dynamics, circuit transferability, and the causes of loss spikes, including those induced by the high curvature of normalization layers. We propose strategies to mitigate these spikes, demonstrating how good visualization facilitates the design of innovative ideas of practical interest. Additionally, we believe our sandbox could assist theoreticians in assessing essential training dynamics mechanisms and integrating them into future theories.

## 1 Introduction

The scaling hypothesis in deep learning states that a model's performance will continue to improve as the size of the model and its training set increase (Hoffmann et al., 2022; Kaplan et al., 2020). This idea has led to a focus on developing very large models, with significant resources dedicated to their training. While this approach has yielded great empirical successes, it requires a substantial computational budget. This makes it challenging to conduct meaningful ablation studies to analyze the importance of different components or factors in a system. As a result, there is still a limited understanding of the *training dynamics* of these large models and much of the knowledge in this area is based on unpublished tricks and techniques rather than rigorous scientific investigation.

We believe that improving the understanding of this matter is of great importance not only to enhance performance but also to limit the failure cases that occur when scaling the size of the models (Chowdhery et al., 2024; Dehghani et al., 2023; Molybog et al., 2023). This may notably contribute to the reduction of memory and budget costs, along with carbon emissions associated with training large models (Faiz et al., 2024). In this paper, we deliberately choose to focus on a small model to gain a deeper understanding of its training dynamics. In particular, the crux of our approach is to consider a model small enough to *visualize it fully*. Specifically, we focus on a transformer architecture with an embedding dimension $d = 2$, which allows us to plot the dynamics of each layer in a two-dimensional plane. This approach enables us to analyze the model's internal workings and visualize the underlying mechanisms that drive its behavior, which may be obscured in larger models. We hope that insights gained from small-scale studies could provide a foundation for understanding the complex dynamics at play in larger models, ultimately shedding light on their behavior and performance.

Our main contribution is to provide a visual sandbox which can be of great interest for several reasons. Theoreticians can use it to build intuition on key mechanisms underlying training dynamics, potentially leading to new theorems. Practitioners can use it to test new ideas, such as optimizer modifications. In particular, modifications to the architecture, to the training setting, or to the data can be seamlessly integrated into our pipeline. By providing this sandbox, we aim to facilitate the understanding of the complex dynamics at play in larger models.

**Summary of our contributions.**   Our contributions are as follows:

- We develop a visual sandbox to visualize the training dynamics of transformers. Our code is made available to the reviewers and will be open-sourced. It is designed to create videos that help comprehend the model's internal dynamics thoroughly.

- Using our visual sandbox, we provide insights into the training dynamics of our model. We classify the types of circuits learned for our problem and illustrate a two-phase learning process, consisting of representation learning followed by classifier fitting.
- We detail the transferability of circuits to showcase the usefulness of curriculum learning and data curation.
- We investigate loss spikes, suggesting potential strategies for mitigation, which could lead to more stable training processes.

Overall, we hope that our study will guide theorists towards new theorems that capture important aspects beyond the training dynamics of transformers and help practitioners build intuition on potentially promising changes to current training pipelines.

**Related Work.**   Our work bridges explainability and training dynamics studies.

*Opening the Black Box.* As neural networks increasingly permeate civil society, concerns about their opacity intensify. This has fueled the ongoing quest to "open the black box." To tackle this issue, researchers have developed various methods, such as extracting meaningful features from neural network activations (Fel et al., 2023), and assessing the impact of perturbations on model inputs (Fel et al., 2021; Koh & Liang, 2017), among others. Recently, the field of mechanistic interpretability has advocated for exposing the internal mechanisms of transformers to provide novel insights into their capabilities (Elhage et al., 2021; Olsson et al., 2022). While some methodologies may apply to large language models (Templeton et al., 2024), precise ablation studies often focus on controlled environments and simplified architectures (Cabannes et al., 2024a; Charton, 2022; Geva et al., 2023; Liu et al., 2022; Meng et al., 2022; Nanda et al., 2023; Rauker et al., 2023; Wang et al., 2023). Our work aligns with this field, as we aim to make the internal behavior of transformers more explicit through carefully selected visualizations in controlled settings. In particular, by utilizing low-dimensional embeddings, we effectively circumvent the curse of dimensionality as defined by Olah (2023). Although low-dimensional embeddings make learning the task more challenging, they enable comprehensive visualization of every model component. However, unlike mechanistic interpretability, our goal is to provide insights into the training dynamics of these models, in the spirit of Wortsman et al. (2024).

*Training Dynamics of Neural Networks.* With the advent of deep learning, new optimization challenges have emerged. Training loss functions are no longer necessarily convex or smooth, the optimization landscape has expanded enormously, and computations are distributed to manage this scale. As the scale of models increases, the absence of theoretical results and closed-form solutions for optimizing neural networks underscores the need to better understand their behavior in practice during training, which is increasingly becoming a computational bottleneck. One approach to address this issue is to employ mathematical abstractions, as seen with neural tangent kernels (NTK) Chizat et al. (2020); Jacot et al. (2018) and mean-field analysis (Chen et al., 2024; Mei et al., 2018), among others (Abbe et al., 2022; Ahn et al., 2024; Cabannes et al., 2024b). Unfortunately, obtaining formal, rigorous results often requires simplifications that deviate from practical implementations. This type of analysis can overlook crucial details that significantly alter the training dynamics.

## 2  SANDBOX SETUP

In this section, we describe our controlled setup, focusing on both the data and the architecture.

### 2.1  SPARSE MODULAR ADDITION

In terms of data, we concentrate on a straightforward mathematical task. We draw inspiration from the sparse parity problem (Abbe et al., 2023; Barak et al., 2023) and the modular addition problem (Nanda et al., 2023; Power et al., 2022) to formulate the *sparse modular addition* problem. This problem is characterized by the following parameters:

- Input length $L \in \mathbb{N}$,
- Vocabulary size $p \in \mathbb{N}$,
- Sparsity index $k \in [L]$, and a set of indices $I \subset [L]$ with cardinality $|I| = k$.

Our default configuration is set to $L = 12$, $k = 5$, and $p \in \{2, 3\}$. Without loss of generality, we assume $I = [k] := \{1, 2, \ldots, k\}$. Inputs are sequences of $L$ tokens $x_t$ in $\mathbb{F}_p = \mathbb{Z}/p\mathbb{Z} \simeq [p]$, and the corresponding targets are the sum of the first $k$ terms modulo $p$. Formally, we aim to learn a mapping:

$$f^* : \quad \begin{aligned} \mathbb{F}_p^L &\to \mathbb{F}_p \\ x = (x_1, x_2, \ldots, x_L) &\mapsto y = \sum_{t \in [k]} x_t. \end{aligned} \qquad \text{(objective)}$$

This mapping defines deterministic conditional distributions linking input and output data through the formula $p(Y = y | X = x) = \mathbf{1}_{\{f^*(x) = y\}}$.

The sparse modular addition task offers the benefits of simplicity while allowing for the observation of a variety of training behaviors.

## 2.2 ONE-LAYER TRANSFORMER ARCHITECTURE

This paper focuses on a specific architecture designed to address our problem.

**Sequence Embeddings.** The sparse modular addition problem is inherently discrete. To handle it with a differentiable architecture, we need to embed it in a continuous space $\mathbb{R}^d$. We first embed each token with both semantic and positional information. Given a learnable token embedding $E : \mathbb{F}_p \to \mathbb{R}^d$, and a learnable position embedding $P : [L] \to \mathbb{R}^d$, a sentence in token space is lifted to a sentence in embedding space through the following embedding operation

$$\forall t \in [L], \qquad z_t := Z(x_t, t) := \frac{E(x_t) + P(t)}{\|E(x_t) + P(t)\|}. \qquad \text{(embedding)}$$

This type of embedding, known as absolute position embedding, incorporates both the semantic meaning of the token and its position within the sequence. The normalization, which projects each element of the sentence onto the sphere, is known as RMS-norm (Zhang & Sennrich, 2019).

Next, an attention mechanism is applied to the sequence of normalized embeddings $(z_t)$ to aggregate them into a single sentence embedding $\xi \in \mathbb{R}^d$. It utilizes a query vector $q \in \mathbb{R}^d$, and a value matrix $V \in \mathbb{R}^{d \times d}$. Denoting $z = (z_t) \in \mathbb{R}^{d \times L}$, the sentence embedding can be expressed in matrix form as

$$\xi := (Vz) \, \text{softmax}\left(\frac{z^\top q}{\sqrt{d}}\right) \in \mathbb{R}^d, \qquad \text{(sentence embedding)}$$

where the softmax operation is defined as a mapping from $\mathbb{R}^L$ to the simplex $\Delta_L$ in $\mathbb{R}^L$. Specifically, the $s$-th component of $\text{softmax}((a_t))$ is proportional to $\exp(a_s)$, formally expressed as $\text{softmax}((a_t)_{t \in [L]}) = (\exp(a_s) / \sum_{t \in [L]} \exp(a_t))_{s \in [L]}$. Compared to the attention mechanism in Vaswani et al. (2023), we omit both the key and output matrices, which would act as extra parameters that do not increase the expressivity of our model.

**Feedforward Neural Network.** Finally, we transform the sentence embeddings using a neural network to cluster them according to the desired output classes. We employ a two-layer multi-layer perceptron (MLP) with pre-norm (Xiong et al., 2020) and residual connection (He et al., 2015). This network is parameterized by $h \in \mathbb{N}$ "receptors" weights $w_i \in \mathbb{R}^d$, $h$ "bias" terms $b_i \in \mathbb{R}$, and $h$ "assemblers" vectors $u_i \in \mathbb{R}^d$ for $i \in [h]$. It implements the following transformation:

$$\zeta := \xi + \sum_{i \in [h]} \sigma\left(\frac{w_i^\top \xi}{\|\xi\|} + b_i\right) \cdot u_i \in \mathbb{R}^d, \qquad \text{(embedding transform)}$$

where $\sigma : \mathbb{R} \to \mathbb{R}$ is a non-linear function, chosen to be $\sigma(x) = x\varphi(x)$ with $\varphi$ being the Gaussian cumulative function. This function $\sigma$ is known as the GELU activation. The final embedding vector $\zeta$ is decoded back to token space based on how it aligns with the respective token embeddings. A softmax layer converts these alignments into a probability vector over the different output classes:

$$p_\zeta(y) = \text{softmax}((E(y)^\top \zeta)_{y \in \mathbb{F}_p}). \qquad \text{(decoding)}$$

Abstracting all the learnable weights into a single parameter $\theta$, our architecture takes as input a sentence $x \in \mathbb{F}_p^L$ and outputs a probability vector over the classes $p_\theta(y|x)$. The parameters of our model are optimized by minimizing the cross-entropy loss defined as:

$$\mathcal{L}(\theta) := \mathbb{E}_{(X,Y) \sim p}\left[-\log(p_\theta(Y|X))\right], \qquad \text{(loss)}$$

where $p$ is a training distribution, typically chosen as the counting measure over some training data. This loss is a proxy for the measure we aim to optimize, which is accuracy, defined as:

$$\mathcal{L}(\theta) := \mathbb{E}_{(X,Y)\sim p}\left[\mathbf{1}_{\{Y=\arg\max_y p_\theta(y|X)\}}\right], \tag{accuracy}$$

with $p$ denoting a data distribution, typically chosen as the counting measure over some testing data.

**Default Configuration.** In our experiments, we trained our networks using $n = 2048 = 2^{11}$ data points, which were sampled uniformly with replacement from the $p^L$ possible sentences. We utilized the Adam optimizer with parameters $\beta_1 = 0.9$ and $\beta_2 = 0.999$ (Kingma & Ba, 2017), and we initialized the network weights using the default schemes provided by PyTorch (Paszke et al., 2019).

## 2.3 VISUALIZATION TOOLS

The primary motivation of this paper is to embed all computations in dimension $d = 2$, enabling us to visualize everything occurring within our model in a relatively straightforward manner. Our codebase is designed to generate videos of the training dynamics, tracking several key aspects. We notably track the following.

**Position Embeddings.** We visualize $P(t)$ for $t \in [L]$ as a point cloud. In this visualization, "spurious" embeddings, $P(t)$ for $t \notin [k]$, are represented by squares, and "non-spurious" ones, $P(t)$ for $t \in [k]$, by circles. Ideally, the transformer would collapse all spurious (resp. non-spurious) position embeddings into a single point, learning invariance of $y$ to sentence suffixes (resp. to non-spurious token position permutation).

**(Normalized) Embeddings.** We visualize the normalized version $Z(x, t)$ of $E(x) + P(t)$ for $(x, t) \in \mathbb{F}_p \times [L]$. We maintain the same circle and square distinction and use the same color for both $E(x) + P(t)$ and $E(x') + P(t)$, for $x' \in \mathbb{F}_p \setminus \{x\}$. On the normalized plot, we also plot the query $q$ as an arrow in $\mathbb{R}^2$, helping us understand how the attention learned where to focus.

**Attention Map.** A concatenation of attention vectors for different sentences is represented as a vectorized image. This visualization enables us to follow the change in activation patterns, even though it is a pure function of the normed embedding visualization.

**Value transform.** We visualize $VZ(x, t)$ as a point cloud. This allows us to understand how sequence embeddings are built and how the value matrix may overcome faulty attention patterns.

**Sequence embeddings/transforms.** We visualize the sequence embeddings $\xi$ (or their transforms $\zeta$) for a set of predefined sentences. These sentences are built by iterating over prefixes $(x_t)_{t \in [k]}$ and suffixes $(x_t)_{t \notin [k]}$. Sentences that share the same prefix have the same color. Sentences whose prefixes are equivalent up to a token position permutation have similar colors. Squares, circles, and triangles are used to distinguish between the classes of the sentences. The sequence embeddings visualization is a direct function of the value transformation and the normalized embeddings.

**Transform level lines.** We visualize the mapping from sentence embedding $\xi$ to their associated learned probabilities $p_\zeta(y = 1)$. We also plot the sentence embedding on the same plot to better understand the level line changes.

**MLP receptors (and assemblers).** We visualize the $w_i \in \mathbb{R}^2$ (and $u_i \in \mathbb{R}^2$) defining the MLP transform as a point cloud in $\mathbb{R}^2$. A consistent color scheme is used to link receptors with the corresponding assemblers.

**Loss and accuracy.** We visualize the current train/test loss and accuracy. These are classical quantities to track. It is interesting to put them in relation to the other visualizations to better understand the loss spikes, loss plateaus, and phase transitions.

## 3 FAMILY OF CIRCUITS

In this section, we describe idealized circuits designed to solve the sparse modular addition problem and discuss concrete solutions implemented by transformers.

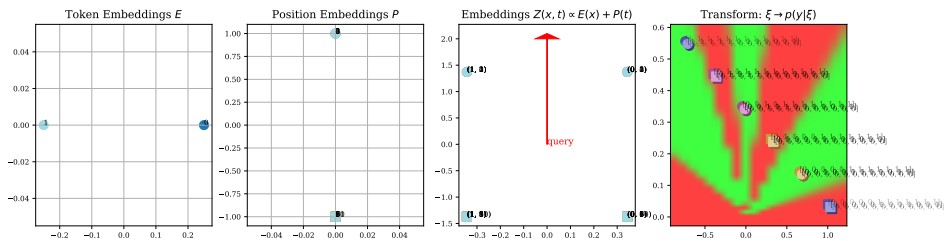

**Figure 1:** Idealized circuit capturing the invariants of the problem

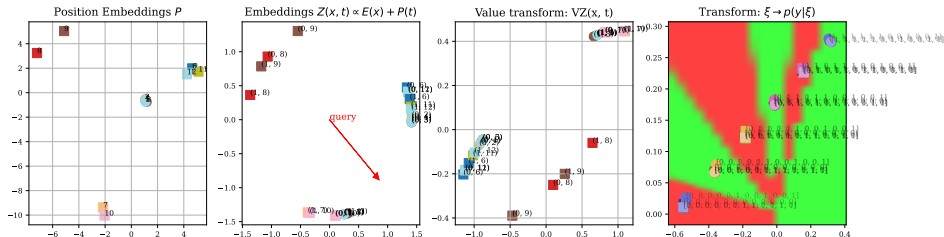

**Figure 2:** Faulty attention corrected by value collapse.

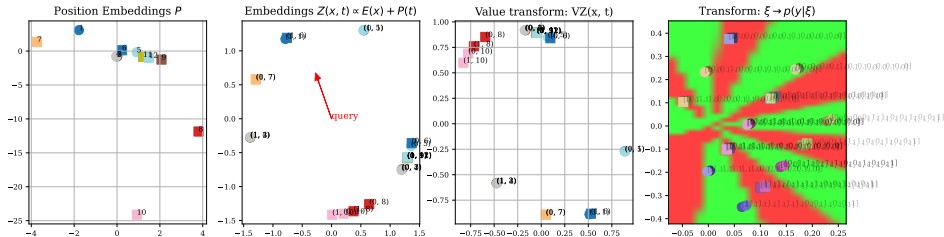

**Figure 3:** Embeddings having only learned some invariants.

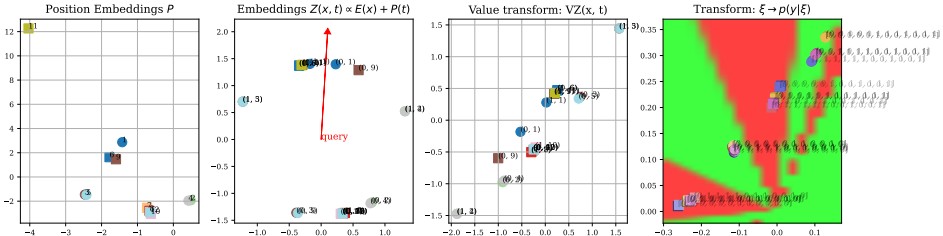

**Figure 4:** Fuzzy construction.

## 3.1 IDEALIZED CIRCUITS

In the sparse modular addition problem, the output $y$ is invariant to two sets of transformations of the inputs $x = (x_1, \ldots, x_L)$.

- "Permutation invariance:" $y$ does not depend on permutation of non-spurious tokens, i.e., if $\pi \in \mathfrak{S}_k$ is a permutation, then $y = f^*((x_i)) = f^*((x_{\pi(1)}, \ldots, x_{\pi(k)}, x_{k+1}, \ldots, x_L))$.
- "Suffix invariance:" $y$ does not depend on the suffix $(x_{k+1}, \ldots, x_L)$.

These two sets of invariants can be easily enforced by the embedding layer. Any architecture where the position embeddings satisfy $p_t = p_1$ for $t \in [k]$, will be permutation invariant, meaning that its output will be invariant to permutations of the non-spurious tokens. Similarly, suffix invariance can be enforced by ensuring that the query vector primarily aligns with $Z(x, t)$ for $t \in [k]$, allowing the sequence embedding $\xi$ to be invariant to the sequence suffix. Such a construction would yield $\binom{k+p-1}{k}$ clusters of sequence embeddings,[1] which the feedforward layer could scatter into as many decision regions to map each sequence embedding $\xi$ to the correct output class $y \in [p]$.

---

[1]This number corresponds to the number of ways to split $k$ into $p$ buckets, which is also the number of stars and bars configurations with $k$ stars and $p-1$ bars.

Figure 1 illustrates this ideal model with our visualization. From left to right, we first plot the two token embeddings $E(x)$ for $x \in [p]$ with $p = 2$. We then plot the twelve position embeddings $P(t)$. The spurious (resp. non-spurious) positions are represented with squares (resp. circles), they all collapse to a single point. The normalized embeddings $Z(x, t)$ are plotted in the third frame, annotated with $(x, t)$, with the query vector $q$ represented as a red arrow. This arrow points in the direction of the embedding $Z(x, t)$ for $t \in [k]$, allowing the attention mechanism to focus exclusively on non-spurious tokens. Finally, we plot some sequence embeddings, annotated with $(x_t)$, and the output of the feedforward transform. The feedforward layer is able to map each cluster to the appropriate output class. Its output is a probability vector over the classes, which we map to a color according to the RGB color wheel.

## 3.2 CONCRETE REALIZATIONS

In practice, the model weights learned through gradient descent deviate from the idealized model. We observe several variations of the idealized model, which we categorize as follows:

**Faulty Attention, Corrected by Value.** In many instances, we find that the attention scores are not fully concentrated on the first five tokens. However, this faulty attention pattern is compensated by the value matrix, which effectively collapses the embeddings of the spurious tokens that are attended to. This adjustment allows the sequence embeddings to remain invariant to the suffix of the sequence. An example of such a configuration is depicted in Figure 2. We also observe examples where the attention focuses solely on non-spurious tokens that are not, for example, zeros, which still allows the sequence embedding to encode for the prefix sums.

**Partially Learned Invariants.** Frequently, we observe that the sequence embeddings have not fully learned all the suffix and prefix invariants, resulting in more than six clusters of sequence embeddings. Specifically, Figure 3 illustrates a scenario where the sequence embeddings are not invariant to the value $x_6$, as evidenced by the positions of the blue squares $(0, 6)$ and $(1, 6)$ on the plot. They also lack invariance to permutations of the token in the first or fifth positions with another of the non-spurious tokens. This results in a sequence embedding that presents more clusters than the idealized model, leading to a greater number of connected decision regions in the feedforward layer.

**Fuzzy Constructions.** Occasionally, we encounter fuzzy constructions where the sequence embeddings are clustered according to unconventional patterns that nonetheless generalize to unseen data. Such a construction is presented in Figure 4.

To conclude, we observe numerous variations from the idealized construction. The networks discover a variety of weight configurations to effectively solve our task. While these configurations do not exhibit particularly striking geometric structures, they often capture the invariants of our problem.

## 4 TRAINING INSIGHTS

This section discusses several training insights we obtained from our sandbox. It focuses on training dynamics, the transferability of circuits, and loss spikes.

### 4.1 TRAINING DYNAMICS

Our toolbox enables us to precisely track the training dynamics of the model. We notably observe that the loss curves typically present drops, corresponding to the learning of different parts of the network, hierarchically from the first to the last layers. They are illustrated in Figure 5.

**First loss drop: learning of the sequence embeddings.** The first loss drop coincides with the learning of the sequence embeddings. It really corresponds to a phase change in the dynamics of the weights. Before the first phase change, the weights seem to wander as if trapped in a saddle point, waiting for a clear signal to escape it. At one point, they all move quite rapidly to create a relatively definitive structure for the sequence embeddings. Interestingly, we also notice that when changing the training hyperparameters, the time for this phase change to occur can vary quite a lot, reflecting the highly unpredictable time needed to escape from the saddle point. Figure 6 shows the high variability

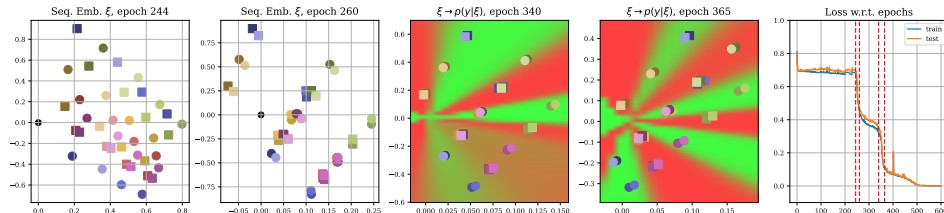

**Figure 5:** *Two-phase learning.* **Right:** Loss profile featuring two significant drops in loss, marked by four red dashed lines at key snapshots. **From left to right: (1)** During the first snapshot, the sequence embeddings lack any clear structure. **(2)** They suddenly become clustered after the first loss drops, as seen in the second snapshot. **(3)** At this point, the MLP already classifies some clusters correctly (third snapshot). **(4)** A second loss drop occurs as the MLP gets fitted (last snapshot).

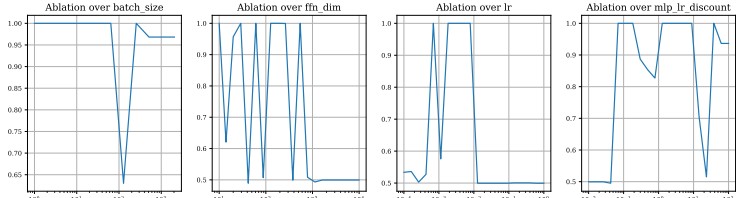

**Figure 6:** *Ablation studies* regarding test accuracy as a function of batch size, hidden neurons, learning rates, and MLP learning rates discount factor for a single run with 1000 epochs. We ensured consistency in initial weights and batch designs when changing hyperparameters.

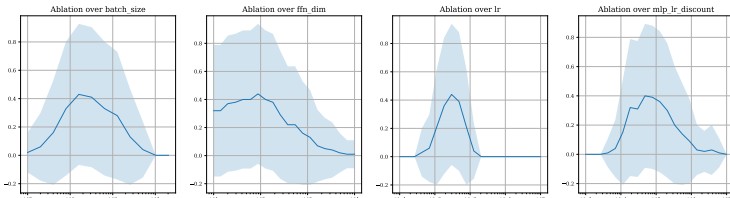

**Figure 7:** This study, akin to Figure 6, yet averaged over 100 runs, highlights some regularity in the effect of hyperparameters on the resulting test accuracy. The solid color indicates standard deviations.

of the test accuracy after 1000 epochs when slightly varying one hyperparameter (Figure 7 shows more regularity when averaging over the runs). The exit from a saddle point can also be understood via the gradient norms as illustrated in Figures 15 and 16 of Appendix A.1. Our findings resonate with some theoretical arguments found in the deep learning theory literature (Chi et al., 2019; Dauphin et al., 2014; Du et al., 2017), results that we hope our toolbox could help theorists strengthen.

**Second loss drop: fitting of the MLP.** The second loss drop is due to the learning of the feedforward network. This change is about fitting the MLP weights to assign the correct classes to the different clusters created during the learning of the sequence embeddings. Interestingly, this second loss drop does not correspond to a clear phase change in the dynamics of the weights. The MLP weights seem to evolve at a continuous speed, although the corresponding decision frontiers change relatively strongly. We notice that this second loss drop appears soon after the first one, if not simultaneously. Once again, we hope that theorists could use our visual toolbox to shed more light on this hierarchical learning of the weights and strengthen previous results on the matter (see, e.g., Abbe et al., 2021).

**The influence of initialization.** As we have seen in the previous subsection, the final configuration can vary quite a lot from one run to another. From our manual inspection, the final configuration seems to be highly correlated with the initial weight configurations. We notably see a strong similarity between the attention patterns at the start and at the end of the training, as illustrated by Figure 8. We also notice that modifying the training hyperparameters (batch size, learning rates, etc.) without changing the initial weights does not significantly alter the final weight configurations.

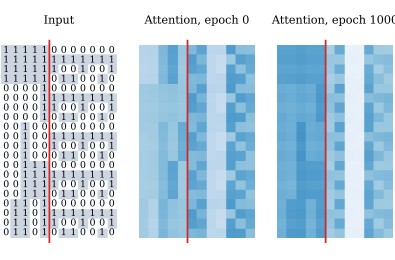 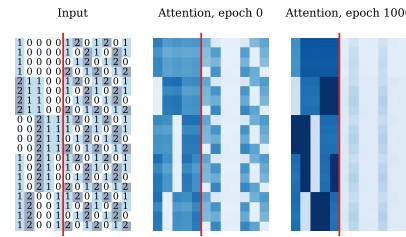

**(a)** Pretraining with $p = 2$.          **(b)** Finetuning with $p = 3$.

**Figure 8:** *Influence of the initialization.* Visualization of attention maps: for each sequence on the left, we plot the corresponding attention pattern at the start and the end of training. The final attention maps are correlated with the original one, illustrating that final variations of the original circuit depend on the original weight configuration.

## 4.2 TRANSFER EXPERIMENTS

Among the lessons learned from training very large models is the importance of careful data engineering (Team, 2024; Touvron et al., 2023b). Cabannes et al. (2024a) has highlighted that certain sources of data may facilitate the learning of invariances, while Abbe et al. (2023) discusses how data curation enables models to escape saddle points more quickly. These insights are consistent with the observations we made regarding our problem. In the sparse modular addition problem, the number of unique sequences is equal to $p^T$, which increases rapidly with both $p$ and $T$ and makes the problem quite hard to learn. In particular, when setting $T = 12$ and limiting the training set to $n = 2048$ data points, training for 1000 epochs does not result in any learning for $p > 4$.

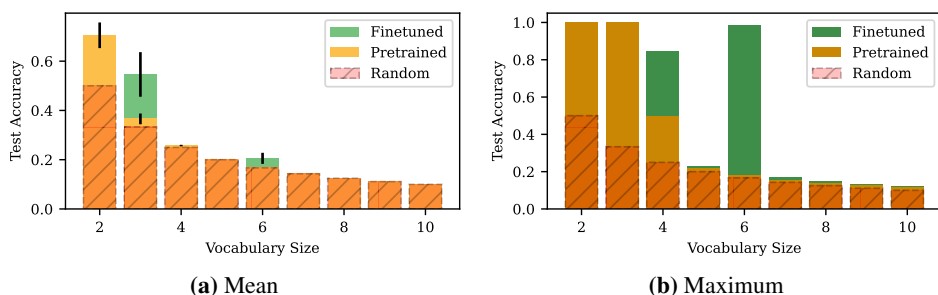

**(a)** Mean                 **(b)** Maximum

**Figure 9:** Accuracies obtained from pretraining only with $p \in [2, 10]$ and finetuning with $p \in [3, 10]$ starting from $p = 2$ for various vocabulary sizes. **Left:** Averaged accuracy; **Right:** Maximum accuracy. Finetuned models display better performances than pretrained-only models.

As previously seen in Figure 2, when training with $p = 2$, we often find circuits that capture both permutation and suffix invariants. These invariants generalize for any $p \in \mathbb{N}$ when $k$ and $T$ are fixed. Consequently, initializing models with these invariants makes learning the sparse modular addition problem much easier. This observation was made after conducting the following experiments: we first trained a model with sequences in $\mathbb{F}_p$ for $p = 2$ over 1000 epochs, before switching the dataset to sequences in $\mathbb{F}_p$ for $p = 3$ for another 1000 epochs. We found that this procedure significantly facilitates the learning of the sparse modular addition problem for $p = 3$, which we summarize in Figure 9 where each bar plot is obtained by over 250 runs. Remarkably, we found that the only models that achieved 100% test accuracy were those that captured both the token and permutation invariances after the first 1000 epochs. Specifically, these were the models that created six sequence embedding clusters, as shown in Figures 1 and 2, rather than those depicted in Figures 3 or 4.

Figure 11 shows the final circuit found in one of our finetuning experiments. The training was initialized with the circuit in Figure 2, after adding a token embedding to encode for $x = 2$, resulting in Figure 10. The final embeddings are not that far from the initial one, with the transformer having learned to mainly pay attention to the non-spurious tokens that are not equal to 2. It also pays some attention to 0 and 1 in positions $t = 7$ and $t = 10$. However, this faulty attention is corrected by the value matrix. Once again, the final configuration seems somewhat close to the initial one, as shown by the attention pattern reported in Figure 8. This is consistent with the observations made in the previous subsections.

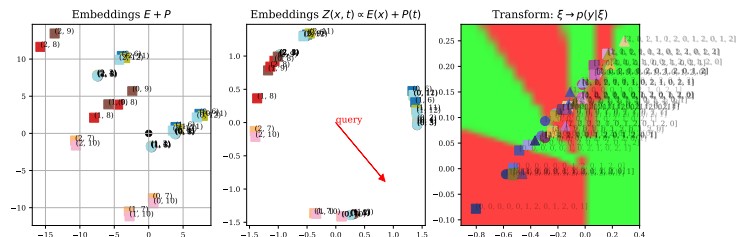

**Figure 10:** Same circuit as in Figure 2 with an additional token embedding for $p = 3$.

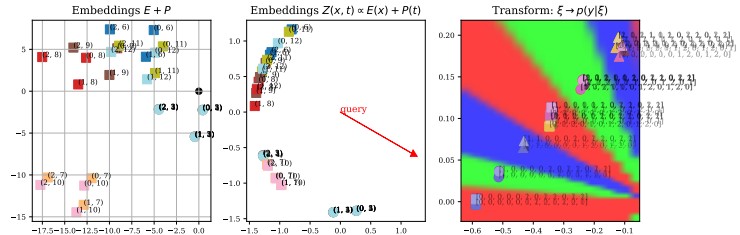

**Figure 11:** Circuit learned after 1000 epochs of finetuning with $p = 3$.

Overall, our transfer experiment highlights the transferability of circuits and the usefulness of a curriculum in facilitating the learning of challenging tasks by inducing effective circuits through tasks that are easier to solve. Although our experiments are performed in two stages, we hypothesize that the same type of mechanism can explain the importance of data curation.

### 4.3 LOSS SPIKES

One interesting aspect of our sandbox is that it generates loss spikes that we can study quite precisely. Our visual inspection showcases two aspects beyond loss spikes. The high curvature of multi-layer perceptrons with heavy weights, or with numerous small correlated weights, as well as the high curvature of the RMS normalization layer near the origin.

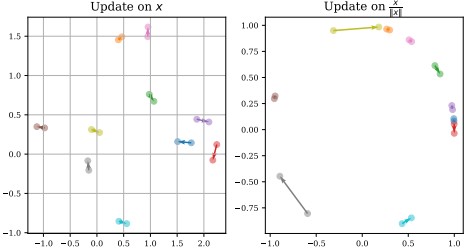

**Figure 12:** Loss spikes are linked to the high curvature of internal network functions. A small update to an element $x$ can result in a substantial change to its normalized version $x / \|x\|$, significantly altering the network's subsequent behavior.

Figure 12 illustrates that a small modification to an element can result in a disproportionately large change in its normalized version. In theory, gradient points towards directions that would reduce the training loss. However, considering a large step size in these directions could be counterproductive. This is especially true for functions with high curvature, such as the normalization layer $f(x) = x / \|x\|$ near the origin. At any point $x_t$, the gradient descent update rule suggests that one can update $x_{t+1}$ as $x_t - \eta_t u_t$ without changing $f(x_t)$, where $u_t = x_t / \|x_t\|$ and $\eta_t$ is the learning rate. This holds true only if the learning is small enough, $\eta_t < \|x_t\|$. When $x_t$ is close to zero, ensuring $\eta_t < \|x_t\|$ becomes challenging, particularly if the step size $\eta_t$ was predetermined by some scheduler. This behavior is related to the "edge-of-stability" phenomenon highlighted by Cohen et al. (2022), further theoretical insights being provided by Cabannes et al. (2024b). Interestingly, this analysis suggests removing some loss spikes by smoothing out the normalization layer. For example, consider using $f(x) = \sigma(\|x\|)x / \|x\|$ where $\sigma$ is a smooth function with $\sigma(0) = 0$ and $\sigma([1, \infty)) = \{1\}$. This demonstrates the usefulness of our visual sandbox in gaining insights and building intuition, which can then be validated on a larger scale in subsequent works.

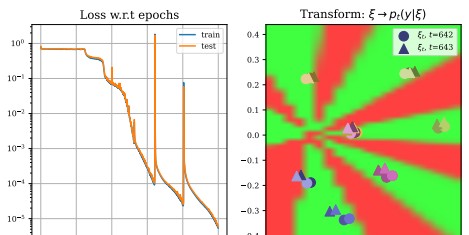 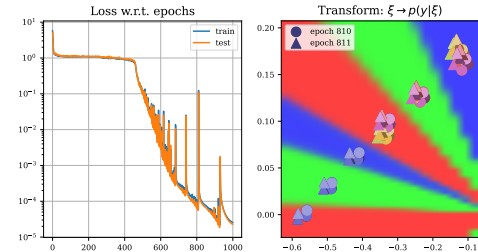

**Figure 13:** Loss spikes (both left) resulting from a small change from one iteration to another in sequence embeddings that are close to the decision boundaries of the subsequent feedforward layer (both right).

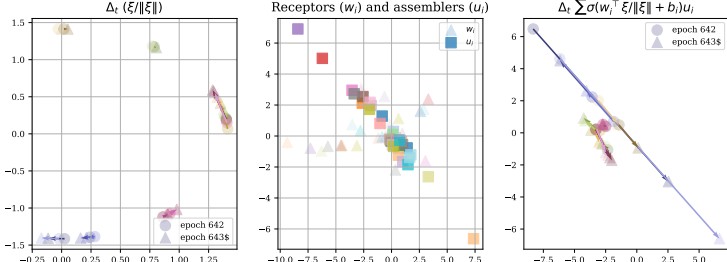

**Figure 14:** A small change in sequence embeddings (left) can lead to a big change in MLP response (right). This is due to heavy, or small but heavily correlated assemblers (middle).

Another source of loss spikes is illustrated in Figure 13. They are due to the decision boundaries of the feedforward layer being quite close to the sequence embeddings, meaning that a small change in the sequence embedding can lead to a high change in their classification. This is again due to the high curvature of the MLP layer, as illustrated in Figure 14. In particular, the heavy, or the small but heavily correlated, weights in the MLP cause the response $\zeta$ to vary highly as a function of $\zeta$. Once again, one can imagine different ways to regularize these types of loss spikes, with various regularization measures, or by ensuring that the capacity of the MLP is large enough for the MLP to avoid creating these heavy or highly correlated weights.

Connection to gradient norms and sparsity is discussed in Appendix A.1 and A.2 and similar experiments with $d > 2$ are conducted in Appendix A.3.

To conclude, our visual sandbox enabled us to peek into some reasons behind loss spikes in large neural networks and to build intuition on how to circumvent them, giving us ideas for large-scale experiments, which, if successful, could help increase the amount of intelligence learned for a given amount of training compute.

## CONCLUSION

This paper aims to advance our understanding of neural network training dynamics through a detailed study of a small-scale transformer model, facilitated by a novel visual sandbox. This tool allows us to observe and analyze the model's internal mechanisms vividly, providing both theoretical and practical insights that could lead to more efficient training strategies.

Our findings highlight a hierarchical learning process where low-level features are developed first, followed by a more predictable model refinement. We also emphasize the impact of initial configurations on the final model outcomes and the seemingly unpredictable nature of some feature learning phase transitions during training. Additionally, we explored the transferability of learned behaviors, which relates to the importance of data curation and curriculum learning in enhancing model performance. We also addressed loss spikes caused by high curvature in the model's internal functions and proposed potential solutions to mitigate these issues. Future work will aim to apply these insights to larger and more complex models, assess the scalability of observed phenomena, and enhance our visualization tools for higher-dimensional models.

**Societal Impact.** This paper aims to understand the complex dynamics at play in neural networks. The insights gained from our research stream could help in designing more efficient and robust neural networks, ultimately aiding in reducing the computational cost and environmental impact of training large models.

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

# A APPENDIX

## A.1 UNDERSTANDING SADDLE POINTS IN LOSS OPTIMIZATION VIA GRADIENT NORMS

When the model's training stagnates, the loss plateaus for several epochs before decreasing again. This behavior may be understood by the optimizer reaching a saddle point in the loss function w.r.t. the model's parameter, that is the gradient of the loss is (almost) zero, but the optimization has not yet reached a local minimum. We illustrate this phenomenon in the full-batch and mini-batch setup, respectively in Figures 15 and 16. The connection between gradient norms and learning phases is salient in Figure 15. We can see that loss drops occur in tandem with high gradient norms for each layer. As studied in Section 4.1, these drops correspond to successive learning phases. This is even more salient in the last subfigure of Figure 15 where the pics precisely match the drops. This shows the connection between the exit of a saddle point and high gradient norms. Similarly, in Figure 16, we can see that the first drop in the loss (around epoch 250 in the first plot) appears when the first increase in gradient norm occurs and the second point of inflection (around epoch 700) also match an infection point on the gradient norms. It should be noted that this is less salient compared to the full-batch setup.

However, this setup enables us to study another training behavior mentioned in Section 4.3. Indeed, we observe in Figure 16 that the loss spikes appear in tandem with high gradient norms, indicating that when a too-large step size deviates the model from its current small loss region, it is taken back to where it was with large updates. It has been shown in the literature (Foret et al., 2021; Ilbert et al., 2024; Zhang et al., 2024) that studying the gradient and the hessian of the loss could provide valuable insights on the neural network optimization both from a theoretical and experimental point of view. We believe using our visual sandbox could help in future work on such an investigation.

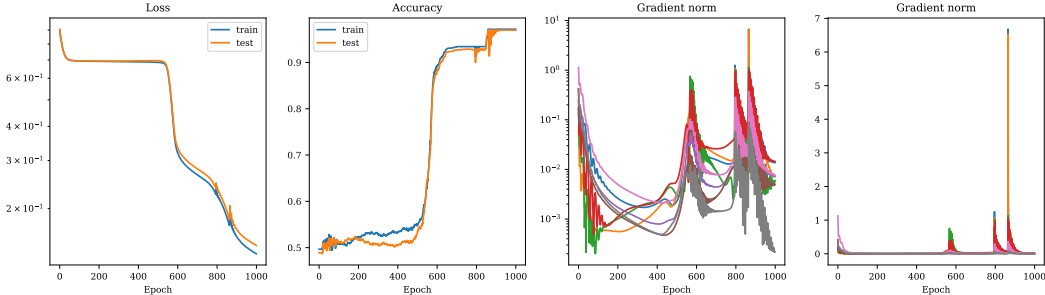

**Figure 15:** *Connection between loss profiles and gradient norm in full-batch setup.* **From left to right:** Evolution of train and test losses, the corresponding accuracies, the evolution of gradient norms for each layer in log-scale, and the similar evolution in linear scale. We see that the learning phases of Section 4.1 appear in tandem with high gradient norms. This can be seen in the last subfigure where the three pics correspond to the loss drops and their corresponding plateaus

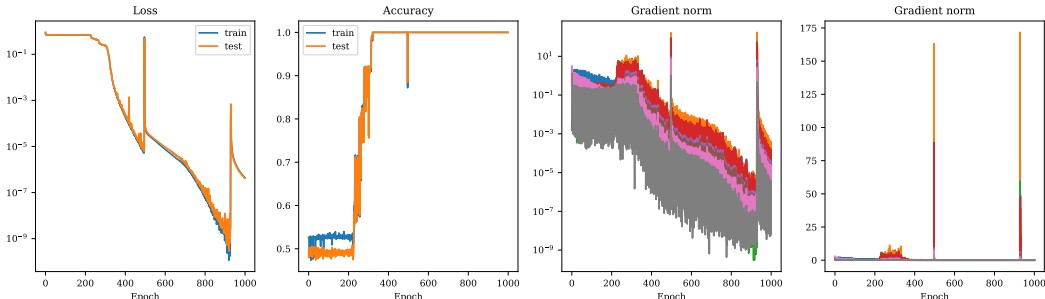

**Figure 16:** *Mini-batch setup.* This study, akin to Figure 15, makes the connection between gradient norm and loss profile even more salient for our problem. The first loss drop appears simultaneously with the first increase in gradient norms. Loss spikes occur in tandem with spikes in gradient norms.

## A.2 Impact of Activation Sparsity on Models Performance

Sparsity is a phenomenon of interest in many fields such as signal processing, neuroscience, and machine learning (Barth & Poulet, 2012; Chen et al., 1998; Mairal et al., 2009). Recent studies focused on the sparsity in deep neural network activations. In particular, Li et al. (2023) showed that trained transformers have sparse activations and concluded that it was caused by the training dynamics rather than by a compact representation of the training data as commonly thought in computer vision and NLP. Mirzadeh et al. (2024) observed a similar phenomenon and showed how to leverage sparsity to reduce the inference cost of large language models. Inspired by this line of work, we study the activation sparsity of our model from a performance viewpoint. It should be noted that those works study deep transformers and identified that the sparsity increases with the depth while we only consider a one-layer transformer.

Following the framework from Li et al. (2023), we recall that the activation sparsity corresponds to the percentage of non-zero entries of the feed-forward activation map. Without loss of generality, the feed-forward block is an MLP with weights $W_1, W_2$ and a non-linear activation $\sigma$ that outputs for any input $x$ a vector $z = W_1\sigma(W_2x)$. Formally, the activation sparsity is the percentage of non-zero entries in $\sigma(W_2x)$ and takes values in $[0, 1]$. In the classical setting with ReLU activation (Fukushima, 1969), this is equivalent to computing the percentage of non-negative neurons before the activation. However, some activations do not have non-negative outputs. This is the case of the SiLU (Elfwing et al., 2018), used in Llama models (Touvron et al., 2023a), and of the GeLU Hendrycks & Gimpel (2023) used in Falcon (Almazrouei et al., 2023), PaLM (Chowdhery et al., 2024), and in our transformer implementation. Instead of replacing such activations by a ReLU (Li et al., 2023; Mirzadeh et al., 2024), we compute a smoothed sparsity[1] as the percentage of entries with an absolute value lower than $\varepsilon > 0$. A sparsity of 1 means that all entries are $\varepsilon$-close to 0 (i.e., sparse activations) and a sparsity of 0 means that all entries are at least $\varepsilon$-away from 0 (i.e., dense activations).

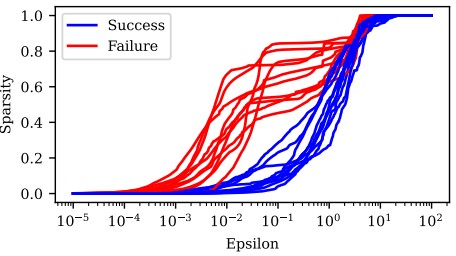 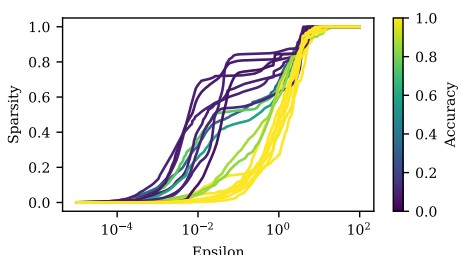

**Figure 17:** *Connection between performance and sparsity.* We display the evolution of the activation sparsity of 20 trained models with $\varepsilon \in [10^{-5}, 10^2]$. **Left:** Successful models (i.e., with test accuracy above 0.9) in blue have less sparse activation than failed models in red. **Right:** The color indicates the models' test accuracy (the lighter, the better). The performance increases as the activation sparsity decreases.

To better understand the impact of sparsity, we train 20 independent models and display in Figure 17 their sparsity after training for $\varepsilon \in [10^{-5}, 10^2]$ (the range is chosen such that the sparsity reaches its extremal values 0 and 1). Given the task's difficulty, achieving an accuracy above 0.9 is a success; otherwise, it is a failure. On the left, we plot successful models in blue and failed ones in red. We observe a striking separation between successful and failed training. In the permissible range $[10^{-3}, 10]$, successful models tend to have less sparse activation than failed ones. To further study this phenomenon, we plot in the right subplot of Figure 17 the evolution of the sparsity with $\varepsilon$, and here, the color indicates the models' test accuracy (the lighter the color, the better the model). We can see that the sparsity decreases as the performance increases. This explains the sharp transition between failure and success in the left subplot. This experiment seems to indicate that, contrary to images and textual data (Li et al., 2023; Mirzadeh et al., 2024), the sparse modular addition problem needs the involvement of many neurons during inference, and hence requires non-sparse activations.

---

[1]This is akin to using the $\ell_1$-norm, respectively the nuclear norm, instead of the $\ell_0$ quasi-norm, respectively the rank (Gribonval & Nielsen, 2003; Ilbert et al., 2024).

### A.3 BEYOND LOW-DIMENSIONAL EMBEDDINGS: EXPERIMENTS WITH $d > 2$

Relying on our visual sandbox, we studied several phenomena on the training dynamics of neural networks from the learning of the different parts of the network in Section 4.1 to the loss spikes occurring during the optimization in Section 4.3 through the efficiency of transfer learning in Section 4.2. Finally, we analyzed the connection between saddle points (and loss spikes) and gradient norms in Appendix A.1 and the impact of the activation sparsity on the models' performance in Appendix A.2. Our rigorous and detailed study was enabled by the low-dimensional embeddings of our transformer model as it makes it possible to visualize each layer of the network. However, we note that many of the studied behaviors can be analyzed independently of the embedding dimension $d$.

In particular, we extend the experiments of Appendix A.1 with an embedding size $d = 3$ in Figure 18. We obtain similar conclusions than in Appendix A.1 where the loss drops and spikes occur in tandem with high gradient norms for each layer. This is even more salient in Figure 19. This again shows the connection between the exit of a saddle point and high gradient norms as well as the connection between gradient norms and loss spikes. We believe using our visual sandbox could help in future work on such an investigation.

Similarly, we extend the experiments of Apppendix A.2 with an embedding size $d \in \{2, 3, 4, 8, 16, 32\}$ in Figure 20. We first note that the higher the embedding size, the more the model succeeds at the task. Especially, as of $d = 8$, all the models are successful, i.e., they all achieve an accuracy higher than $0.9$ (as defined in Appendix A.2). This is expected given that imposing low-dimensional embeddings limits the expressiveness and the generalization power of our model. It should be noted that this was one of the many challenges of our study: obtaining a generalizable neural network with embeddings in $R^2$ for a mathematical reasoning task such as the sparse modular addition problem. We obtain similar conclusions than in Appendix A.2 with successful models having more dense activations.

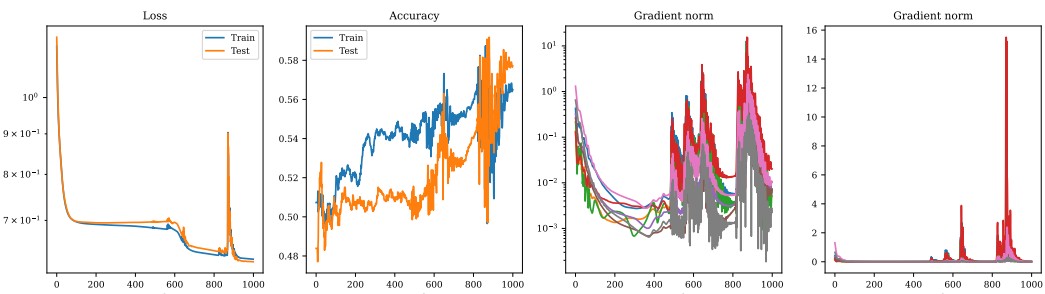

**Figure 18:** *Connection between loss profiles and gradient norm in full-batch setup when $d = 3$.* **From left to right:** Evolution of train and test losses, the corresponding accuracies, the evolution of gradient norms for each layer in log-scale, and the similar evolution in linear scale. Askin to Figure 15, we see that the loss profiles studied in Section 4.1 and Section 4.3 appear in tandem with high gradient norms. This can be seen in the last subfigure where the pics correspond to the loss drops and spikes.

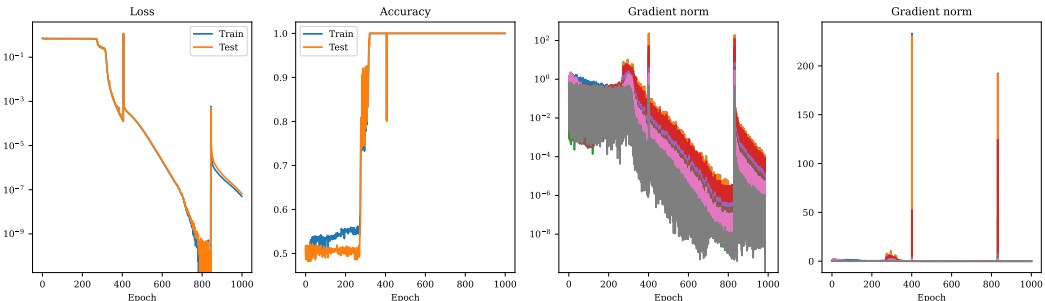

**Figure 19:** *Mini-batch setup.* This study, akin to Figure 18, makes the connection between gradient norm and loss profile even more salient for our problem. The first loss drop appears simultaneously with the first increase in gradient norms. Loss spikes occur in tandem with spikes in gradient norms.

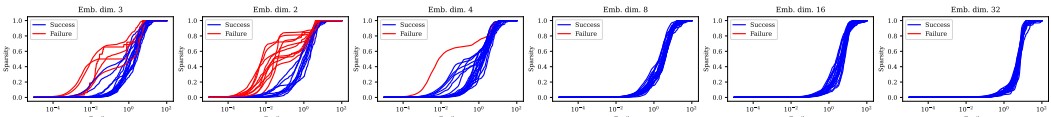

**Figure 20:** *Connection between performance and sparsity.* We display the evolution of the activation sparsity of 20 trained models with $\varepsilon \in [10^{-5}, 10^2]$ for $d \in \{2, 3, 4, 8, 16, 32\}$. Successful models (i.e., with test accuracy above 0.9) in blue have less sparse activation than failed models in red.