# OpenReview forum: "A Visual Case Study of the Training Dynamics in Neural Networks"
_ICLR.cc/2025/Conference — Submitted to ICLR 2025_

### Official Review · Reviewer_1hd3 · 2024-11-03

**Soundness:** 2
**Presentation:** 1
**Contribution:** 2
**Rating:** 3
**Confidence:** 4

**Summary:**

This paper examines the training dynamics of neural networks by conducting a detailed study on a small-scale transformer model, employing a novel visual sandbox. This tool facilitates the observation and analysis of the model's internal mechanisms, offering insights into more efficient training strategies.

**Strengths:**

- The experimental motivation and setup are clearly articulated.

**Weaknesses:**

- The presentation of the experimental results is suboptimal. In Figure 4, many values overlap, resulting in readability issues.
- There is a lack of theoretical support for the proposed method. The paper claims, "This tool allows us to observe and analyze the model’s internal mechanisms vividly, providing both theoretical and practical insights that could lead to more efficient training strategies." However, the theoretical insights remain unclear.
- The experiments are limited. The paper lacks experiments across varied datasets, tasks, and neural network architectures.

**Questions:**

- Could you present the experimental results more clearly?
- Could you provide theoretical support for the proposed method?
- Could you conduct additional experiments across various datasets, tasks, and neural network architectures?

---

> ### Author Response · Authors · 2024-11-24
> **Answer to Reviewer 1hd3**
>
> We thank the Reviewer 1hd3 for their valuable comments. We are happy that the reviewer acknowledged the clarity of our setup and problem motivation.
>
> We address the reviewer's concerns below.
>
> **Experimental investigation and theoretical insights**: Our visual sandbox provides insights into the training dynamics of neural networks. Hence, it mostly consists of an experimental yet rigorous study. We however note that in our additional experiments in Appendix A, we study phenomena that are of interest both theoretically and empirically in the literature (loss spikes, gradient norms, activation sparsity) [1, 2, 3, 4, 5].
>
> **Beyond low-dimensional embeddings**: In Appendix A, we conduct additional experiments on loss spikes, gradient norms, and activation sparsity and also extend them to the case $d>2$. We note however that the goal of our study is to remain in a controlled setup (transformer model and modular addition problem) to better understand the training dynamics as is often the case in mechanistic interpretability papers. However, the insights our sandbox enables us to obtain can be generalized to other setups and architectures. In particular, as explained in the revised version of the paper, the experiments of Sections 4.1, 4.2, 4.3, and Appendix A are related to phenomena studied in the literature [1, 2, 3, 4, 5] on large-scale transformers and for various tasks (time series forecasting, computer vision, LLMs).
>
> We thank the reviewer for their time. We hope we addressed their concerns and we would be happy to answer any additional questions.
>
> [1] Li et al. The lazy neuron phenomenon: On the emergence of activation sparsity in transformers. ICLR 2023.
>
> [2] Mirzadeh et al. ReLU strikes back: Exploiting activation sparsity in large language models. ICLR 2024.
>
> [3] Foret et al. Sharpness-aware minimization for efficiently improving generalization. ICLR 2021.
>
> [4] Ilbert et al. SAMformer: Unlocking the potential of transformers in time series forecasting with sharpness-aware minimization and channel-wise attention. ICML 2024
>
> [5] Zhang et al. Why transformers need Adam: A hessian perspective, NeurIPS 2024.

---

> > ### Comment · Reviewer_1hd3 · 2024-11-26
> >
> > Thank you for your response. However, my concerns remain unresolved.
> >
> > (1) Firstly, the motivation for the proposed method remains insufficiently explained, and the proposed method is not grounded in adequate theoretical justification. Although the authors reference an analysis of experimental results in Appendix A, this does not meaningfully address the core of my concern.
> >
> > (2) Additionally, the analysis predominantly focuses on a small-scale transformer model applied to two-dimensional data, which limits its practicality and raises questions about its scalability to more complex, real-world applications.
> >
> > (3) Finally, the presentation of the paper and the clarity of the experimental results require improvement. For example, in Figures 1–4, overlapping values significantly hinder interpretability. I strongly recommend the authors to reorganize the paper’s structure to enhance clarity and coherence. Therefore, I maintain my score.

---

### Official Review · Reviewer_XhQn · 2024-11-04

**Soundness:** 2
**Presentation:** 2
**Contribution:** 2
**Rating:** 3
**Confidence:** 3

**Summary:**

In this work, the authors presented a case report that using 2D visualization to demonstrate the training dynamics of a neural network trained to solve the sparse modular addition problem. Various aspects of the training dynamics, including the two stages of learning, the influence of initialization, process of network adaptation in sequential learning, trajectory of the loss function including its anomaly (the spikes) are visualized and analyzed qualitatively.

**Strengths:**

By choosing a well-structured task and proper visualization, the authors have demonstrated a number of interesting and potentially important phenomena regarding network training dynamics. The framework presented may be extended to study other networks and tasks, leading to better understanding of networks’ learning process.

**Weaknesses:**

It is unclear what part of the current work can generalize. Is it the methodology of 2D embedding and visualization? If so, what is the difficulty of doing it and what is the innovative way to overcome them in the present work? And how the proposed method can generalize to other tasks and networks? These needs to be explained and clarified in a clearer way. Or, is the conclusion of the training dynamics generalizable? But here it seems unlikely, because the current observations were based on a particular network structure trained with a specific task. Even for the current network and task, the authors need to show how robust of the observations, e.g., the two stages of training, with one for embedding and one for classification. Without either the method or the conclusions generalizable to more situations, the impact of the work might be limited.

**Questions:**

I suggest the authors re-organize the paper to better explain and clarify what is the problem solved by this case report, and how it can help readers working in training networks in general.

---

> ### Author Response · Authors · 2024-11-24
> **Answer to Reviewer XhQn**
>
> We thank the reviewer XhQn for their valuable comments. We are happy to read that the reviewer found our findings interesting and potentially important.
>
> We address the reviewer's concerns below.
>
> **Motivation for our controlled setup**: The motivation for focusing on $d=2$ was to be able to visualize all the training processes in $R^2$. While such visualization could be extended to $d=3$, albeit to the cost of simplicity and clarity, it cannot be done for higher $d$ unless involving dimensionality reduction. This explains why we focused on 2D embeddings although our insights extend to more practical settings.
>
> **Insights into more practical settings**: The behaviors identified thanks to our sandbox on the training dynamics can be generalized to a more practical setup. In particular, the efficiency of transfer learning is known in the literature. More importantly, controlling the loss spikes phenomenon is very important to stabilize the training of large models such as LLMs. To address the reviewer's concerns, we extended our study of loss profiles to phenomena studied in the literature on gradient norms and activation sparsity [1, 2, 3, 4, 5] in Appendix A.1 and A.2 and we extended these experiments to the case $d>2$ in Appendix A.3.
>
> We thank the reviewer for their valuable comments. We hope we have addressed their concerns and we would be happy to answer any additional questions.
>
> [1] Li et al. The lazy neuron phenomenon: On the emergence of activation sparsity in transformers. ICLR 2023.
>
> [2] Mirzadeh et al. ReLU strikes back: Exploiting activation sparsity in large language models. ICLR 2024.
>
> [3] Foret et al. Sharpness-aware minimization for efficiently improving generalization. ICLR 2021.
>
> [4] Ilbert et al. SAMformer: Unlocking the potential of transformers in time series forecasting with sharpness-aware minimization and channel-wise attention. ICML 2024
>
> [5] Zhang et al. Why transformers need Adam: A hessian perspective, NeurIPS 2024.

---

> > ### Comment · Reviewer_XhQn · 2024-11-26
> >
> > I appreciate the authors' efforts in addressing my previous comments. However, I still have significant concerns:
> > 1.	It remains unclear what obstacles in visualizing the training dynamics of neural networks have been overcome by the current work. Additionally, how does the proposed method generalize to different network structures and tasks?
> > 2.	To what extent do the phenomena observed during training provide insights into network training dynamics in general, given that these observations are based on a specific network trained for a particular task?
> > While the newly added analyses are helpful, I am afraid they do not fully address these questions.

---

### Official Review · Reviewer_qxou · 2024-11-04

**Soundness:** 3
**Presentation:** 3
**Contribution:** 3
**Rating:** 6
**Confidence:** 3

**Summary:**

This paper presents a novel visual sandbox designed to explore the training dynamics of a small-scale transformer model with a restricted embedding dimension of $d=2$. This setup allows for two-dimensional visualization, providing a detailed look into the model’s internal mechanisms. The sandbox helps uncover hierarchical learning processes, circuit transferability, and causes of loss spikes, particularly those due to high curvature in normalization layers. The authors propose methods to mitigate these spikes, highlighting how visualization can lead to more effective training strategies. They suggest that this tool could assist in developing theories and improving training processes for more complex neural networks.

**Strengths:**

- The paper introduces a novel tool for visualizing the training dynamics of transformers. The visualization offers both practical and theoretical insights, making complex mechanisms more accessible.

- The authors identify a hierarchical learning process inside a pretrained transformer via their visualization tool, showing how low-level features are established before higher-level refinement. This is valuable for understanding model behavior and training stages.

- The tool also helps to recognize loss spikes caused by high curvature in the model's normalization layers. Based on this, the authors suggest practical mitigation strategies, potentially making training more stable and efficient.

**Weaknesses:**

- My first concern is that, while the visualization tool is insightful, its restriction to small-scale transformers with $d=2$ limits its applicability to larger and more complex models commonly used in practice. It remains unclear whether insights from this sandbox will generalize to high-dimensional transformers. Further discussions on the higher dimensional embedding should be provided.

- There is no comparison between a pretrained model and a fine-tuned model based on insights derived from the sandbox. Consequently, it remains unclear whether the recommendations informed by the sandbox visualizations effectively contribute to improved model performance on practical tasks.

**Questions:**

- Could the authors discuss the scalability of their sandbox tool to higher-dimensional transformer models, for example, when $d=3$? How does the computational complexity of this sandbox grow in terms of $d$?

- Are there quantitative metrics for evaluating the effectiveness of the proposed loss spike mitigation strategies?

- What types of theoretical developments/problems in training dynamics can be recognized or conjectured from the visualization results of this sandbox?

---

> ### Author Response · Authors · 2024-11-24
> **Answer to Reviewer qxou**
>
> We thank the reviewer qxou for their detailed feedback. We appreciate that the reviewer found our work novel and valuable.
>
> We address the reviewer's concerns point by point below.
>
> > 1. My first concern is that, while the visualization tool is insightful, its restriction to small-scale transformers with $d=2$ limits its applicability to larger and more complex models commonly used in practice. It remains unclear whether insights from this sandbox will generalize to high-dimensional transformers. Further discussions on the higher dimensional embedding should be provided.
>
> We thank the reviewer for their suggestion. To address the reviewer's concerns, we added two experiments in Appendix A.1 and A.2 on the connection between gradient norms and loss spikes and the connection between activation sparsity and models' performance. These experiments are connected to the insights presented in our submission and are agnostic to the embedding dimension. As a matter of fact, we extended these two experiments to the case $d>2$ in Appendix A.3.
>
> >2. Could the authors discuss the scalability of their sandbox tool to higher-dimensional transformer models, for example, when $d=3$? How does the computational complexity of this sandbox grow in terms of $d$?
>
> We note that the motivation for focusing on $d=2$ was to be able to visualize all the training processes in $R^2$. While such visualization could be extended to $d=3$, albeit to the cost of simplicity and clarity, it cannot be done for higher $d$ (unless involving dimensionality reduction). However, the training behaviors identified thanks to our visual sandbox can be observed on large-scale models. We especially study phenomena of interest in large-scale models (gradient norms, activation sparsity, loss spikes) [1, 2, 3, 4, 5] in Appendix A.1 and A.2 and went beyond the case $d>2$ in Appendix A.3.
>
>
> >3. Are there quantitative metrics for evaluating the effectiveness of the proposed loss spike mitigation strategies?
>
> Loss spikes occur during the training of models and hence are hard to quantitatively measure or monitor. We show in Appendix A.1 and A.3 (Figures 15, 16, 18, 19) that loss spikes and high gradient norms occur in tandem. Hence, one could think of quantitative metrics on gradient norms to better assess the quality of a loss spikes mitigation strategy. Recently, there has been some proposal of new normalization schemes to stabilize the training of neural networks training [6, 7, 8]. We believe such strategies (or similar ones) could be efficient in controlling the gradients of the layers and hence the loss spikes.
>
> > 4. What types of theoretical developments/problems in training dynamics can be recognized or conjectured from the visualization results of this sandbox?
>
> We acknowledge that the benefits of our sandbox for theoretical problems were not clearly stated. To address the reviewer's concern, we conducted additional experiments motivated by our visual sandbox in Appendix A on phenomena, such as loss spikes, gradient norms, and activation sparsity that are of interest both theoretically and experimentally in the literature on large-scale transformers [1, 2, 3, 4, 5]. We believe our sandbox can help better understand and elucidate such problems using dynamical observations on a small scale before tackling them on bigger models.
>
> We thank the reviewer for their time and feedback. We hope to have addressed their concerns and would be happy to answer any additional questions.
>
> [1] Li et al. The lazy neuron phenomenon: On the emergence of activation sparsity in transformers. ICLR 2023.
>
> [2] Mirzadeh et al. ReLU strikes back: Exploiting activation sparsity in large language models. ICLR 2024.
>
> [3] Foret et al. Sharpness-aware minimization for efficiently improving generalization. ICLR 2021.
>
> [4] Ilbert et al. SAMformer: Unlocking the potential of transformers in time series forecasting with sharpness-aware minimization and channel-wise attention. ICML 2024
>
> [5] Zhang et al. Why transformers need Adam: A hessian perspective, NeurIPS 2024.

---

> > ### Comment · Reviewer_qxou · 2024-11-27
> >
> > Dear Authors! Thank you for your responses, which have helped me understand the paper better. Your replies have addressed my concerns. However, I agree with the other reviewers that the analysis focuses on a small-scale transformer model applied to two-dimensional data with just a slight discussion on high-dimensional cases during rebuttal. This focus limits its practicality and raises questions about its scalability to more complex, real-world applications. Overall, I prefer to maintain my score.

---

> ### Comment · Area_Chair_oW9N · 2024-11-27
> **Rebuttal Response**
>
> Dear Reviewer,
> Do you mind letting the authors know if their rebuttal has addressed your concerns and questions? Thanks!
> -AC

---

> ### Comment · Reviewer_B9Wq · 2024-11-27
>
> Thank you to the authors for their detailed reply. Makes sense to me.

---

### Author Response · Authors · 2024-11-24
**General Comment**

We thank all the reviewers for thoroughly and carefully reading our paper. We are grateful to them for acknowledging the **clarity** of our setup and motivations (Reviewers XhQn,  1hd3) and for finding our study **novel and valuable** to better understand training dynamics (Reviewer qxou).

Several reviewers raised a common concern regarding the takeaways of our study for more practical settings as well as the low dimensional case treated in our study ($d=2$). To clear out any possible misunderstanding regarding the scope of our work, we provide clarifications below.

- **Low-dimensional embeddings:** Our focus on a simplified transformer architecture in low dimension was motivated by the need to be able to visualize all the steps the neural network operates during training. Our visual sandbox leads to detailed and rigorous observations of the training process and thanks to the controlled setup (modular addition problem), one can better interpret the training dynamics.

- **Beyond $d=2$**: Thanks to our visual sandbox, we studied several phenomena on the training dynamics (e.g., layers learning, transfer learning, loss spikes). As explained in our paper, these phenomena are of great interest for the training of large-scale transformer models such as LLMs. Moreover, we note that many of the studied behaviors can be analyzed **independently of the embedding dimension $d$.** To address the reviewers' concerns, we conducted three additional experiments connected to the phenomenon studied in our submission: **(1)** connection between loss profiles and saddle points via gradients norms in Appendix A.1, **(2)** connection between activation sparsity and models' performance in Appendix A.2, **(3)** extension of these 2 experiments for $d > 2$.


- **Practical takeaways**: These experiments are strongly connected to phenomena observed in the literature on large-scale transformer models [1, 2, 3, 4, 5] which makes our study relevant in practical settings. As demonstrated by the research process described in our work, we believe our visual sandbox can enable rigorous study in small-scale scenarios to elucidate and then study them on bigger models. We believe this can be valuable to researchers and practitioners to better understand the training dynamics of neural networks.

**Updates to the paper (in blue)**

We list below the updates to the paper (in blue):
- Added Figures 15, 16, 17, 18, 19,and  20 in Appendix A
- Experiment on gradient norms in Appendix A.1
- Experiment on activation sparsity in Appendix A.2
- Similar experiments with $d>2$ in Appendix A.3

We thank the reviewer for their time and valuable feedback that helped us improve our work. We remain at their disposal to continue this collaborative discussion until the end of the rebuttal.

---
[1] Li et al. The lazy neuron phenomenon: On the emergence of activation sparsity in transformers. ICLR 2023.

[2] Mirzadeh et al. ReLU strikes back: Exploiting activation sparsity in large language models. ICLR 2024.

[3] Foret et al. Sharpness-aware minimization for efficiently improving generalization. ICLR 2021.

[4] Ilbert et al. SAMformer: Unlocking the potential of transformers in time series forecasting with sharpness-aware minimization and channel-wise attention. ICML 2024

[5] Zhang et al. Why transformers need Adam: A hessian perspective, NeurIPS 2024.

---

---

### Meta-Review · Area_Chair_oW9N · 2024-12-16

**Metareview:**

This paper applies various methods for visualizing the training dynamics of a small transformer model with embedding dimension 2, trained on a sparse modular addition problem.  Various aspects of training dynamics are observed and studied including the impact of initialization, observation of two stages of learning, circuit transferability, and loss spikes during training.

Some reviewers felt the paper presented a novel tool for visualizing training dynamics. Other reviewers noted the task and visualization strategies were well chosen to provide insight into the training dynamics.  The visualizations and analysis showed interesting phenomena in the training dynamics inducing a multi-part learning process and loss spikes caused by normalization layers.  These may lead to insights for improving training.

Reviews generally agreed, however, that the embedding dimension was too small to draw strong conclusions about larger models, limiting the applicability to real-world models and problems.  Moreover, it is not readily clear how the method in the paper may even be applied to larger models or other task functions.  Some reviewers also pointed out the method lacked a strong theoretical basis.

Reviewers reached consensus that questions about the practical drawbacks of the method outweighed the demonstrated insights and observations.

**Additional Comments On Reviewer Discussion:**

The reviewers and authors actively engaged in discussion.  In particular, the authors provided additional experiments in the rebuttal showing the method could work for slightly larger embedding dimension.  This did not sway reviewers.  While there was one borderline accept vote, during AC-reviewer discussion reviewers reached consensus that the weaknesses outweighed the strengths.

---

### Decision · Program_Chairs · 2025-01-22

Reject